

# Low-Power WALTZ Decoupling under Magic-Angle Spinning

Luzian Thomas[1] and Matthias Ernst[1]

[1]Department of Chemistry and Applied Biosciences, ETH Zürich, Vladimir-Prelog-Weg 2, 8093 Zürich, Switzerland

**Correspondence:** Matthias Ernst (maer@ethz.ch)

**Abstract.** Heteronuclear low-power decoupling using the solution-state WALTZ sequences has become quite popular in solid-state protein NMR and seems to work well. However, there are no systematic studies that characterize these sequences under magic-angle spinning (MAS) and give recommendations which parameter should be used. We have studied in detail the use of WALTZ-16 and WALTZ-64 as low-power decoupling sequences under 100 kHz MAS by characterizing the resonance conditions analytically, using numerical simulations, and experiments on model substances. The recoupling at heteronuclear resonance conditions between the modulation frequency of the sequences and the MAS frequency is the most important feature. Pulse lengths corresponding to areas with vanishing first-order heteronuclear recoupling are good candidates for efficient decoupling. We have characterized two such conditions defined by $\nu_1 = \nu_r/10$ or $\tau_{90} = \tau_r/4$ and $\nu_1 = 2\nu_r/5$ or $\tau_{90} = \tau_r$ which both lead to narrow lines and are stable against rf-field variations and chemical-shift offsets. More such conditions might exist but were not investigated here.

## 1 Introduction

Heteronuclear spin decoupling (Ernst, 2003; Hodgkinson, 2005) is an essential component for recording high-resolution NMR spectra of nuclei like $^{13}$C or $^{15}$N in solids under magic-angle spinning (MAS). Even at the highest spinning frequencies available today ($\nu_r \approx 160$ kHz (Callon et al., 2023) and beyond), residual dipolar-coupling terms as well as the interplay of splittings due to isotropic $J$ couplings and proton spin diffusion (Sinning et al., 1976; Mehring and Sinning, 1977; Ernst et al., 1998), lead to line broadening. At spinning frequencies below 50 kHz, typically high-power decoupling with nutation frequencies at least three times the spinning frequency are used in order to avoid resonance conditions. At spinning frequencies above 50 kHz, low-power decoupling sequences with $\nu_1 < \nu_r/3$ can be used and are of advantage due to the significantly lower rf-field requirement. Over the years, a number of low-power decoupling sequences have been introduced and characterized. Most of them are sequences first developed for high-power decoupling in solids under MAS and then adapted for low-power use (Ernst et al., 2001, 2003; Griffin et al., 2007; Kotecha et al., 2008; Weingarth et al., 2009; Mithu et al., 2011; Agarwal et al., 2013; Equbal et al., 2017; Simion et al., 2022). They vary mainly in the effort required to optimize the parameters for obtaining as narrow lines as possible.

The line width under decoupling of any heteronuclear decoupling sequence under MAS is given by three contributions (Ernst, 2003): (i) the residual line width under the sequence which is given by the first- and second-order effective non-resonant





Hamiltonian. These are typically residual terms originating from the incomplete averaging of the isotropic heteronuclear $J$ or the heteronuclear dipolar coupling. (ii) Contributions from nearby resonance conditions that lead to a partial reintroduction of the heteronuclear dipolar coupling. (iii) Proton spin diffusion in combination with residual couplings can lead either to line

broadening or to line narrowing (self decoupling) due to exchange broadening or exchange narrowing, respectively, between the two multiplet lines when the proton spin changes its state (Sinning et al., 1976; Mehring and Sinning, 1977; Ernst et al., 1998). The first two points can be characterized by simulations of small spin systems or by effective Hamiltonian calculations while the last point is often difficult to assess using theory or numerical simulations.

Besides the low-power decoupling sequences mentioned above, also composite-pulse sequences originally developed for solution-state NMR spectroscopy have been used at high MAS frequencies. Especially WALTZ-16 (Shaka et al., 1983a, b) and WALTZ-64 (Shaka and Keeler, 1987) have become popular low-power decoupling sequences in solid-state protein NMR spectroscopy (Marchand et al., 2022). They have, however, to the best of our knowledge never been characterized and it is unclear what the optimum parameter conditions are. The WALTZ sequences are fairly long (WALTZ-16 consists of 96

90° pulses and WALTZ-64 of 384 90° pulses), leading to the possibility of many resonance conditions since the modulation frequency $\omega_{\mathrm{m}} = 2\pi/\tau_{\mathrm{m}}$, where $\tau_{\mathrm{m}}$ is the cycle time of the sequence, is fairly small compared to the MAS frequency. However, at high MAS frequencies, resonance conditions will require the condition $n_0\omega_{\mathrm{r}} + k_0\omega_{\mathrm{m}} = 0$ to be fulfilled that requires high values of $k_0$. Typically, Fourier coefficients become smaller with higher values of $k_0$ and it is not clear whether under these conditions the Fourier coefficients are still significant and will lead to line broadening. In order to understand these question

better and to predict for which parameters good decoupling can be obtained, we have analyzed the performance of WALTZ sequences for low-power decoupling using analytical calculations of residual coupling terms and resonance conditions as well as numerical simulations. The theoretical predictions are then verified using experimental measurements at 100 kHz MAS using the methylene group of glycine ethylester as a test substance.

## 2 Methods

### 2.1 Analytical Calculations

Analytical calculations were performed in the framework of operator-based Floquet theory (Scholz et al., 2010; Leskes et al., 2010; Ivanov et al., 2021). The time-dependent Hamiltonian $\mathcal{H}(t)$ due to magic-angle spinning (MAS) is transformed into an interaction frame with the radio-frequency field irradiation by

$$\mathcal{H}'(t) = U_{\mathrm{rf}}^{-1}(t)\mathcal{H}(t)U_{\mathrm{rf}}(t) \tag{1}$$

with

$$U_{\mathrm{rf}}(t) = \hat{T}\exp\left(-i\int\limits_0^t \mathcal{H}\mathrm{rf}(t')dt'\right) \tag{2}$$





where $\hat{T}$ is the Dyson time-ordering operator (Dyson, 1949). The interaction-frame trajectory can be characterized by a Fourier series

$$I'_z(t) = U_{\mathrm{rf}}^{-1}(t) I_z U_{\mathrm{rf}}(t) = a_{zx}(t) I_x + a_{zy}(t) I_y + a_{zz}(t) I_z = \sum_{k=-\infty}^{\infty} \left( a_{zx}^{(k)} I_x + a_{zy}^{(k)} I_y + a_{zz}^{(k)} I_z \right) \tag{3}$$

assuming a cyclic sequence, i.e., $U(\tau_{\mathrm{m}}) = \mathbf{1}$. We obtain, therefore, a Fourier-series representation of the Hamiltonian with two frequencies:

$$\mathcal{H}'(t) = \sum_{n=-2}^{2} \sum_{k} \mathcal{H}'^{(n,k)} e^{in\omega_{\mathrm{r}}t} e^{ik\omega_{\mathrm{m}}t} \tag{4}$$

Detailed expressions for the Fourier coefficients can be found in Eq. (69) in Scholz et al. (2010). Using operator-based Floquet theory, we can calculate the non-resonant contributions in first and second order as

$$\bar{\mathcal{H}}^{(1)} = \mathcal{H}'^{(0,0)} \tag{5}$$

and

$$\bar{\mathcal{H}}^{(2)} = -\frac{1}{2} \sum_{\nu,\kappa} \frac{\left[ \mathcal{H}'^{(-\nu,-\kappa)}, \mathcal{H}'^{(\nu,\kappa)} \right]}{\nu\omega_{\mathrm{r}} + \kappa\omega_{\mathrm{m}}} \tag{6}$$

where the summation goes over all values of $(\nu, \kappa)$ where $\nu\omega_{\mathrm{r}} + \kappa\omega_{\mathrm{m}} \neq 0$ and the modulation frequency $\omega_{\mathrm{m}} = 2\pi/\tau_{\mathrm{m}}$. Detailed expressions for the second-order terms can be found in Eqs. (11-17) in Tan et al. (2016). If the interaction-frame transformation

includes the chemical-shift offset or if the pulse sequence is not cyclic ($U(\tau_{\mathrm{m}}) \neq \mathbf{1}$), the treatment has to be extended to a triple-mode Floquet treatment (Scholz et al., 2010; Tan et al., 2016).

From the resonance conditions $n_0\omega_{\mathrm{r}} + k_0\omega_{\mathrm{m}} = 0$ and the Fourier coefficients obtained from the interaction-frame trajectory (Eq. (3)), we can directly obtain the effective heteronuclear Hamiltonians at the resonance conditions which is given by

$$\bar{\mathcal{H}}^{(1)} = \mathcal{H}'^{(n_0,k_0)} = \omega_{IS}^{(n_0)} S_z \left( a_{zx}^{(k_0)} I_x + a_{zy}^{(k_0)} I_y + a_{zz}^{(k_0)} I_z \right) = \omega_{IS}^{(n_0)} a_z^{(k_0)} S_z I_{z'} \tag{7}$$

The Fourier coefficients of the interaction-frame trajectory $a_z^{(k_0)} = \sqrt{\left( a_{zx}^{(k_0)} \right)^2 + \left( a_{zy}^{(k_0)} \right)^2 + \left( a_{zz}^{(k_0)} \right)^2}$ encode the scaling factor of the heteronuclear dipolar coupling at the resonance conditions. We can, therefore, use them directly as a relative measure for the strength of the recoupling.

For a given $\omega_1 = -\gamma B_1$ rf-field amplitude, we can calculate the modulation frequency of the WALTZ sequences as $\omega_{\mathrm{m}}^{(\mathrm{W})} = \omega_1/z_0$ where $z_0 = 24$ and $z_0 = 96$ for WALTZ-16 and WALTZ-64, respectively. Based on the resonance condition $n_0\omega_{\mathrm{r}} +$

$k_0\omega_{\mathrm{m}} = 0$, we can, therefore, calculate the rf-field amplitude for each resonance condition characterized by $k_0$ leading to $\omega_1 = z_0 n_0 \omega_{\mathrm{r}}/k_0$. Using the rf-field amplitude corresponding to the resonance condition, we can then calculate the Fourier coefficients that describes the scaling factor for the residual dipolar coupling (see Eq. (7)).

The second-order contributions are typically given by the commutator terms of Eq. (6) and are either cross terms between the heteronuclear dipolar coupling and the I-spin CSA tensor or the heteronuclear dipolar coupling and the homonuclear dipolar





coupling. In principle, the magnitude of these terms can also be calculated analytically (Tan et al., 2016). Sequences like the WALTZ sequences without an effective field show only contributions from the dipole-dipole cross term while the dipole-CSA cross term is zero due to symmetries in the Fourier coefficients of the interaction-frame trajectory. If the chemical-shift offsets are included into the interaction-frame trajectory, both kinds of cross terms are usually contributing to the second-order residual line width. In this paper, we will not calculate the second-order terms analytically but rely on numerical simulations.

In two-spin simulations with only the heteronuclear dipolar coupling, only first-order resonance effects will play a role while in three-spin systems with all interactions also second-order terms become relevant. While a realistic simulation of decoupling efficiency requires prohibitively large spin systems, simulations in a $CH_2$ group can capture all resonant and non-resonant features originating from first- and second-order contributions.

## 2.2 Numerical Simulations

Numerical simulations for small model spin systems were carried out using the spin-simulation environment GAMMA (Smith et al., 1994). Isolated two-spin systems with just a dipolar coupling with an anisotropy of $\delta_{IS}/(2\pi) = $ -45.3 kHz were used to characterize the resonance conditions. In addition, two-spin systems with an additional CSA tensor on the irradiated I spin as well as $I_2S$ three-spin systems with $J$ couplings, CSA tensor, and homonuclear dipolar coupling were simulated for a better characterization of the decoupling performance of the WALTZ sequences. The details of the parameters used in the numerical

simulations can be found in the SI, section S1. All numerical simulations were processed with an exponential apodization of 30 Hz to avoid truncation artifacts in the spectra. Therefore, all simulated intensities are relative to a perfectly decoupled line with a processing line width of 30 Hz.

## 2.3 Experimental Measurements

All measurements were performed on a Bruker Avance NEO 850 MHz spectrometer equipped with a double-resonance 0.7 mm

MAS probe spinning at 100 kHz. For each measurement, four scans with 6144 data points and a spectral width of 52631 Hz were added up. Adiabatic cross polarization from protons to $^{13}C$ was used to increase the sensitivity of the measurements followed by an acquisition on $^{13}C$ under proton decoupling using the WALTZ sequences with variable parameters. The rf-field amplitudes were calibrated using a nutation experiment at several power levels and linear extrapolation of the power levels to the required range of amplitudes. Only very minor deviations of the nutation frequency from the requested power levels were

found. For the measurement of the offset dependence, the carrier frequency was switched to the required value before the start of the acquisition. All decoupling sequences were implemented as asynchronous composite-pulse decoupling sequences and no effort was made to synchronize the sequences with the MAS rotation. As a sample, the $CH_2$ group of $^{15}N$-1,2-$^{13}C$ glycine ethyl ester was used. All measurements were recorded in a single measurement session. Throughout the measurement session, the intensity of 1D spectra was checked to ensure a stable setup over the whole time.



## 3 Results and Discussion

In the main part of the paper we will discuss the results based on analytical calculations, numerical simulations and experimental data using the WALTZ-64 sequence for decoupling. The equivalent figures for WALTZ-16 decoupling can be found in the SI, section S2, where all the figures of the main paper are duplicated showing the results for WALTZ-16. The cycle time of WALTZ-16 is only a quarter of the cycle time of WALTZ-64 which means that the corresponding values of $k_0$ are also a quarter of the ones for WALTZ-64. Therefore, the spacing of the resonance condition is a factor of four larger. This implies that the difference of the pulse length between two adjacent points in the analytical calculations and the numerical simulations is by a factor of four larger in WALTZ-16 compared to WALTZ-64.

### 3.1 Resonance Conditions

Analytical calculations of the strength of the resonance conditions were done for a MAS frequency of 100 kHz. The Fourier coefficients $a_z^{(k_0)}$ were calculated from an interaction-frame trajectory as a function of the value of

$$k_0 = -n_0 \frac{\omega_{\mathrm{r}}}{\omega_{\mathrm{m}}} = -n_0 z_0 \frac{\omega_{\mathrm{r}}}{\omega_1} = -n_0 4 z_0 \frac{\tau_{90}}{\tau_r} \tag{8}$$

where $\omega_{\mathrm{r}}$ is the MAS frequency, $\omega_{\mathrm{m}}$ the modulation frequency of the pulse sequence, $\omega_1$ the nutation frequency of the rf-field amplitude, $\tau_{90}$ the pulse length of the basic 90° pulse in the WALTZ sequence, $\tau_{\mathrm{r}}$ the length of a rotor periode, and $z_0$ the number of $2\pi$ rotations in the WALTZ sequences. For WALTZ-16, $z_0 = 24$, while for WALTZ-64, $z_0 = 96$. The strength of the residual coupling on a resonance condition is given by $a_z^{(k_0)}$ (see Eq. (7)) and is plotted in Fig. 1 for WALTZ-64 as a function of $k_0$. One can clearly see that even for very large values of $k_0$ the Fourier coefficients still have a significant magnitude on the order of $10^{-3}$ considering the magnitude of typical one-bond $^1$H-$^{13}$C or $^1$H-$^{15}$N couplings with $\delta_{\mathrm{D}}/(2\pi) \approx$ -46 and 20 kHz, respectively. Assuming a ratio of $\omega_{\mathrm{r}}/\omega_1 = 10$, the resonance conditions would be characterized by $k_0 = 960$ for $n_0 = 1$ and $k_0 = 1920$ for $n_0 = 2$ (Fig. 1b and c). Using the above mentioned typical value of $10^{-3}$ for the Fourier coefficients leads to a residual couplings in the order of 20-50 Hz. However, there are clearly ranges of values of $k_0$ where the Fourier coefficients are zero over a larger range as can be seen in Fig. 1b around $k_0 = 960$ or $k_0 = 930$ where the first one corresponds to $\omega_{\mathrm{r}}/\omega_1 = 10$. There are many more areas like this, e.g., at $\omega_{\mathrm{r}}/\omega_1 = 4$, corresponding to $k_0 = 384$ (Fig. 1d and e) but we will only investigate the two mentioned here. All of these areas with zero Fourier coefficients could be good ranges for stable and efficient decoupling using WALTZ-64 since the resonant contribution to the effective Hamiltonian is zero. Simulations for WALTZ-16 (see SI, Fig. S1) show very similar results but since the cycle time is shorter, fewer almost zero Fourier coefficients are found adjacent to each other. On the other hand, the spacing of the resonance conditions is larger by a factor of four, as mentioned above, due to the larger modulation frequency (shorter cycle time) of WALTZ-16 compared to WALTZ-64.

The resonance conditions can also be assessed in numerical simulations of two-spin systems where only the heteronuclear dipolar coupling is included. In such a system, no higher-order terms exist and the line intensity is directly a measure for the residual coupling generated by the resonance conditions. Figure 2 shows a plot of the line intensity in a C-H two-spin system at 100 kHz MAS with a heteronuclear dipolar coupling anisotropy of $\delta_{\mathrm{CH}}/(2\pi)$ = -45 kHz as a function of the 90° pulse length.



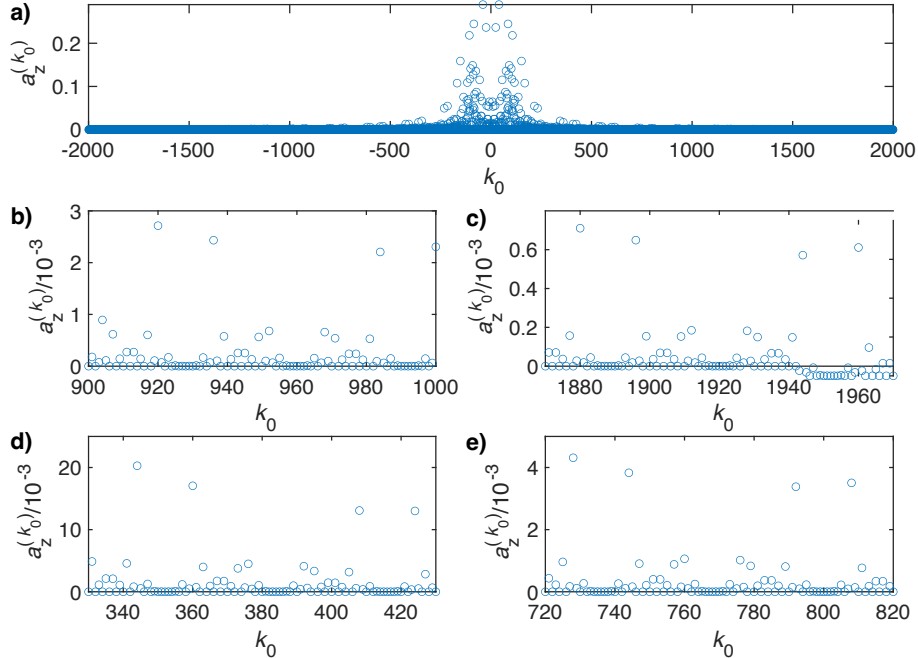

**Figure 1.** Plot of the Fourier coefficients $a_z^{(k_0)}$ as a function of $k_0$ for the WALTZ-64 pulse sequence assuming ideal rectangular pulses. One can clearly see that even for very high values of $k_0$ many of the resonance condition still have a significant contribution. a) Complete range from $k_0 = -2000$ to 2000. b) Enlarged range around $k_0 = 960$ corresponding to a ratio of $\omega_\mathrm{r}/\omega_1 = 10$ for $n_0 = 1$. c) Enlarged range around $k_0 = 1920$ corresponding to a ratio of $\omega_\mathrm{r}/\omega_1 = 10$ for $n_0 = 2$. d) Enlarged range around $k_0 = 384$ corresponding to a ratio of $\omega_\mathrm{r}/\omega_1 = 4$ for $n_0 = 1$. c) Enlarged range around $k_0 = 768$ corresponding to a ratio of $\omega_\mathrm{r}/\omega_1 = 4$ for $n_0 = 2$.

The relationship between $\tau_{90}$ and $k_0$ is given by Eq. (8). One can clearly see many resonance conditions up to pulse length of $\tau_{90} = 50$ μs (Fig. 2) corresponding to an rf-field amplitude of 5 kHz. Expanded areas around a pulse length of 25 μs (10 kHz rf-field amplitude, Fig. 2b) and 10 μs (25 kHz rf-field amplitude, Fig. 2c) show a similar pattern as in the analytical calculations in Fig. 1. There are ranges of pulse lengths (corresponding to specific values of $k_0$) where no line broadening due to resonance conditions are observed and good decoupling might be possible.

Besides the first-order resonance conditions discussed above, non-resonant first-order contributions as well as second-order cross terms between the heteronuclear dipolar coupling and either the proton CSA tensor or homonuclear dipolar couplings on the protons determine the residual line width under decoupling. To assess how big these contributions are, we simulated $CH_2$ three-spin systems including $J$ couplings, proton CSA tensors, and proton homonuclear dipolar couplings (for the simulation parameters see the SI, section S1. Figure 3 shows the peak height as a function of the pulse length $\tau_{90}$ at 100 kHz MAS for WALTZ-64 decoupling. The obtained line intensities are quite similar to the ones shown in Fig. 2 except that the intensity outside the resonance conditions is slightly reduced compared to an ideal two-spin C-H system. This indicates that there is some second-order broadening coming mostly from the second-order cross term between the heteronuclear and homonuclear





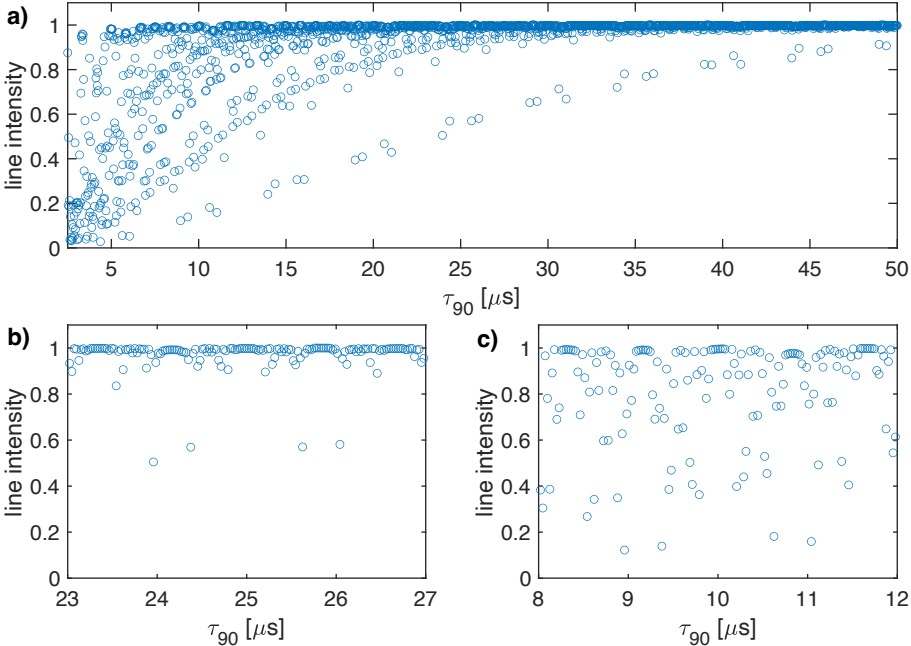

**Figure 2.** Plot of the simulated line intensity in a two-spin system with only the heteronuclear dipolar coupling as a function of the pulse length $\tau_{90}$ for the WALTZ-64 pulse sequence assuming ideal rectangular pulses. The corresponding rf-field amplitude for each value of $\tau_{90}$ can be calculated as $\nu_1 = 1/(4\tau_{90})$. The pulse length correspond to values of $\tau_{90} = k_0\tau_r/(4*z_0)$. a) Complete range from $\tau_{90} = 2.5$ to 50 μs corresponding to $k_0 = 96$ to 1920. b) Enlarged range around $\tau_{90} = 25$ μs corresponding to an ideal $B_1$ field of 10 kHz. c) Enlarged range around $\tau_{90} = 10$ μs corresponding to an ideal $B_1$ field of 25 kHz. As described in the materials and methods section, all simulations are processed with an exponential apodization of 30 Hz with an intensity of one corresponding to a non decaying line.

160   dipolar coupling as can be seen from Fig. S11 of the SI which shows simulations of a CH two-spin system including the proton CSA tensor that shows no significant second-order broadening compared to Fig. 2 .

We can also characterize the resonance conditions experimentally by simultaneously incrementing the pulse length of the basic 90° pulse of the WALTZ sequence while adjusting the rf-field amplitude such that the flip angle of the pulses remains constant. Figure 4 shows a plot of the line intensity of the $C_\alpha$ carbon resonance (CH$_2$ group) in $^{15}$N-1,2-$^{13}$C glycine ethyl ester

165   as a function of the pulse length $\tau_{90}$ for the WALTZ-64 pulse sequence at 100 kHz MAS. The line intensity is normalized to the maximum intensity obtained using optimized high-power ($\nu_1 = 250$ kHz) XiX decoupling (Tekely et al., 1994; Detken et al., 2002). The resonance conditions agree quite well with the ones found in the simulations (see Fig. 3) and we find again that there is an area without significant line broadening originating from resonance conditions around $\tau_{90} = 25$ μs ($\nu_1 \approx 10$ kHz, see Fig. 4b ) and around $\tau_{90} = 10$ μs ($\nu_1 \approx 25$ kHz, see Fig. 4c). A similar plot for WALTZ-16 decoupling can be found in the

170   SI, Fig. S4. Keep in mind that the experimental data was measured on a full rotor subject to rf-field inhomogeneity that was not included in the numerical simulations. Therefore, the resonance conditions are not as sharp as in the numerical simulations and show a convolution of the rf-field distribution with the pattern of the resonance conditions. In contrast to the simulations (Figs.





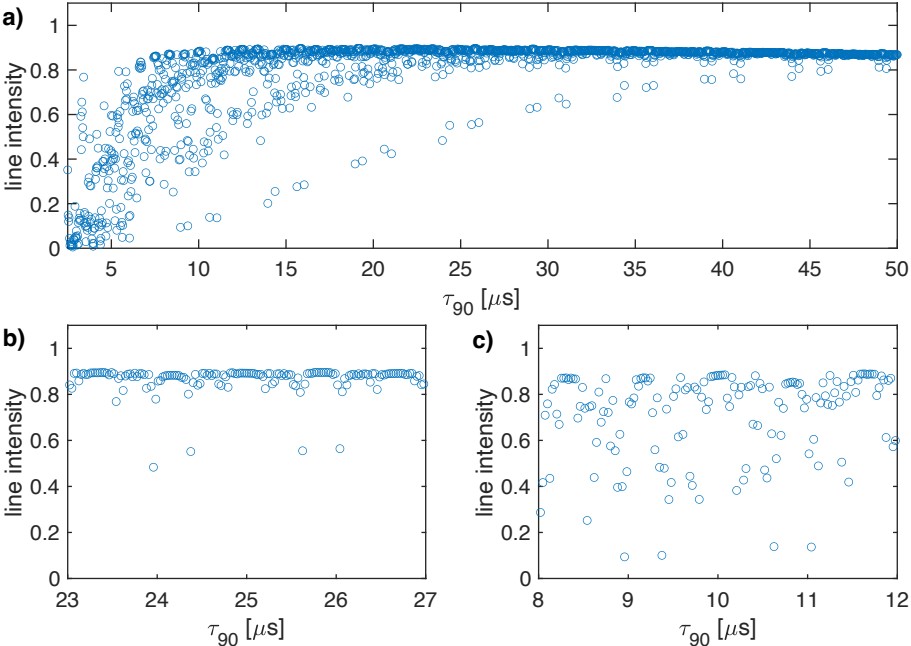

**Figure 3.** Plot of the simulated line intensity in a $CH_2$ three-spin system with dipolar couplings and CSA tensors as a function of the pulse length $\tau_{90}$ for the WALTZ-64 pulse sequence assuming ideal rectangular pulses. The corresponding rf-field amplitude for each value of $\tau_{90}$ can be calculated as $\nu_1 = 1/(4\tau_{90})$. The pulse length correspond to values of $\tau_{90} = k_0\tau_r/(4 * z_0)$ a) Complete range from $\tau_{90} = 2.5$ to 50 μs corresponding to $k_0 = 96$ to 1920. b) Enlarged range around $\tau_{90} = 25$ μs corresponding to an ideal $B_1$ field of 10 kHz. c) Enlarged range around $\tau_{90} = 10$ μs corresponding to an ideal $B_1$ field of 25 kHz. As described in the materials and methods section, all simulations are processed with an exponential apodization of 30 Hz with an intensity of one corresponding to a non decaying line.

2 and 3), there is a significant decay of the overall observed line intensity with increasing pulse length. This can be attributed to higher-order effects that become bigger for longer cycle times of the WALTZ sequence. In addition, also decoupling side
bands at multiples of the modulation frequency will increase in intensity for longer cycle times (Sachleben et al., 1996) and reduce the observed center-band intensity.

    Based on the analytical calculations, numerical simulations, and the experimental data, we can conclude that there are many potential areas for good low-power decoupling using the WALTZ sequences. We have investigated two pulse length that correspond to rf-field amplitudes $\omega_1 = \omega_r/10$ and $\omega_1 = \omega_r/4$ in more detail since they seemed to be the most promising ones
based on the experimental data. While the first one shows slightly lower intensities than the second one, the closest significant resonance condition is further away which might be of advantage. In a second step, we will analyze how stable the two areas are against missettings of the rf-field amplitude by the experimentalist or due to a distribution of rf-field amplitudes in the coil (rf-field inhomogeneity) that are always present in experiments.





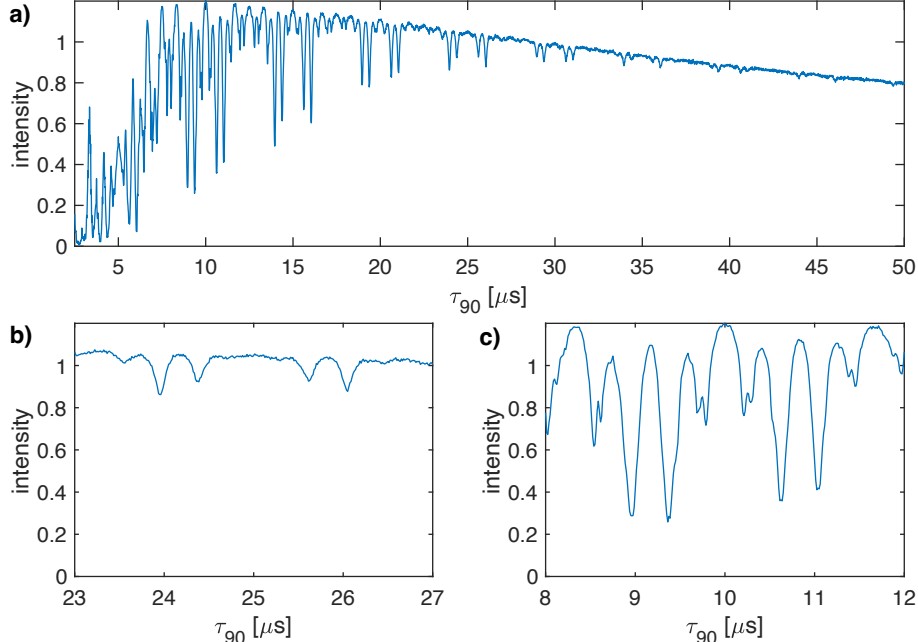

**Figure 4.** Plot of the measured line intensity of the $CH_2$ group in 1,2-$^{13}$C glycine ethyl ester as a function of the pulse length $\tau_{90}$ for the WALTZ-64 pulse sequence at 100 kHz MAS. The rf-field amplitude was adjusted such that the flip angle was always 90°. The increment of the pulse length was set to 12.5 ns. a) Complete range from $\tau_{90}$ = 2.5 to 50 μs. b) Enlarged range around $\tau_{90}$ = 25 μs corresponding to a $B_1$ field around 10 kHz. c) Enlarged range around $\tau_{90}$ = 10 μs corresponding to a $B_1$ field around 25 kHz. While the numerical calculations and simulations of Figs. 1-3 could be carried out exactly on the resonance conditions, this was not possible for the experimental data due to the limited time resolution of the pulse programmer and the limited stability of the MAS rotation. Therefore, the experimental data are displayed as a line plot with the highest possible time resolution.

## 3.2 Stability against rf-field amplitude changes

In the analytical calculations of the resonance conditions, we can change the rf-field amplitude while keeping the timing constant to see how the resonance conditions that depend only on the timing of the WALTZ sequences change as a function of $B_1$. Stability against variations in the $B_1$ field are important for two reasons. Firstly, experimentally, $B_1$ fields are often determined in a simple way using the zero crossing of a 180° pulse at high power and are then extrapolated to lower powers assuming a linear amplitude scale. While modern spectrometers have often linearized frequency generation and amplification pathways,

the determination of the $B_1$ fields has still a quite large error margin. Secondly, as discussed above for the experimental determination of the resonance conditions, the sample will always experience a distribution of rf-field amplitudes since different points inside the coil will experience different $B_1$ fields. Therefore, it is important that the resonance conditions and the areas without resonance conditions used for decoupling are at the same location if the $B_1$ changes.



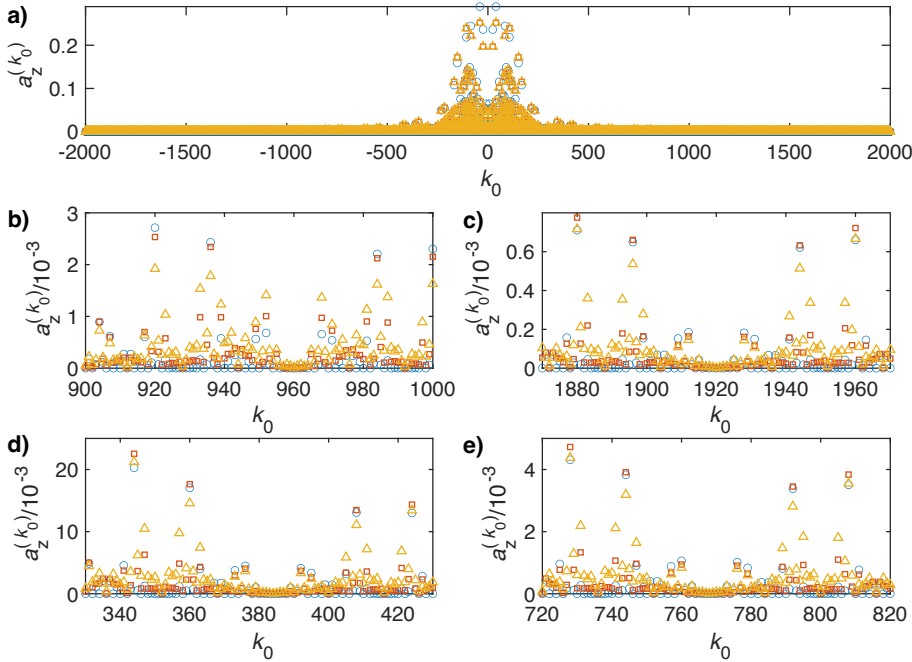

**Figure 5.** Plot of the Fourier coefficients $a_z^{(k_0)}$ as a function of $k_0$ for the WALTZ-64 pulse sequence assuming ideal rectangular pulses. Blue circles correspond to the ideal $B_1$ field (same data as Fig. 1), red squares to a $B_1$ field that is increased by 10% and yellow triangles to one that is increased by 20%. a) Complete range from $k_0 = -2000$ to 2000. b) Enlarged range around $k_0 = 960$ corresponding to a ratio of $\omega_r/\omega_1 = 10$ for $n_0 = 1$. c) Enlarged range around $k_0 = 1920$ corresponding to a ratio of $\omega_r/\omega_1 = 10$ for $n_0 = 2$. d) Enlarged range around $k_0 = 384$ corresponding to a ratio of $\omega_r/\omega_1 = 4$ for $n_0 = 1$. c) Enlarged range around $k_0 = 768$ corresponding to a ratio of $\omega_r/\omega_1 = 4$ for $n_0 = 2$.

Figure 5 shows the strength of the resonance conditions ($a_z^{(k_0)}$) in WALTZ-64 decoupling for an ideal $B_1$ field (blue circles),
a $B_1$ field that is 10% larger (red squares) and a $B_1$ field that is 20% larger (yellow triangles) compared to the theoretical value. An equivalent plot for $B_1$ field amplitudes 10 and 20% too low can be found in the SI, Fig. S14. One can clearly see that some of the resonance conditions broaden and that some of the areas with almost zero Fourier coefficients (see Fig. 1 for comparison) show now significant intensity. However the areas corresponding to $\omega_r/\omega_1 = 10$ ($k_0 = 960$ for $n_0 = 1$ and $k_0 = 1920$ for $n_0 = 2$ in Fig. 5b and c, respectively) and $\omega_r/\omega_1 = 4$ ($k_0 = 384$ for $n_0 = 1$ and $k_0 = 768$ for $n_0 = 2$ in Fig. 5d and e, respectively) show for all three rf-field amplitudes only small resonant contributions. This indicates that these two areas should be stable against mis settings of the rf-field amplitude as well as against rf-field inhomogeneity.

A more detailed analysis of the sensitivity of resonance conditions to timing and rf-field deviations for the two regions of interest ($\omega_r/\omega_1 = 10$, $k_0 = 960$ for $n_0 = 1$, $k_0 = 1920$ for $n_0 = 2$ and $\omega_r/\omega_1 = 4$, $k_0 = 384$ for $n_0 = 1$, $k_0 = 768$ for $n_0 = 2$) was obtained by independently changing the length of the pulse and the amplitude of the rf field in analytical and numerical calculations. Figure 6 shows both regions of interest ($\omega_r/\omega_1 = 10$ on the left, Fig. 6a and c, and $\omega_r/\omega_1 = 4$ on the right, Fig.



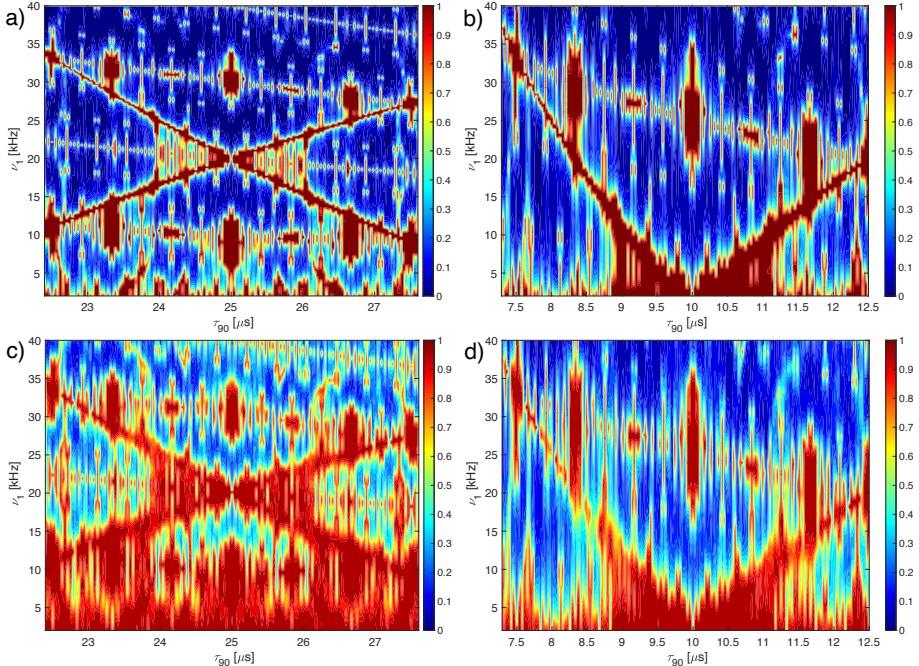

**Figure 6.** Plot of the line intensities as a function of the pulse length $\tau_{90}$ and the rf-field amplitude $\nu_1$ for the WALTZ-64 pulse sequence assuming ideal rectangular pulses around the conditions $\omega_r/\omega_1 = 10$ (left side) and $\omega_r/\omega_1 = 4$ (right side). The topmost row shows the analytical calculations of the resonance intensity a) around $k_0 = 960$, $\nu_1 = 10$ kHz, $\tau_{90} = 25$ µs, b) around $k_0 = 384$, $\nu_1 = 25$ kHz, $\tau_{90} = 10$ µs. The residual coupling was converted into a Gaussian line with the corresponding line width and the line intensity of this line is plotted to make the plots easier comparable to the numerical simulations. The second row shows the peak height of the numerical simulations for a two-spin system with only a heteronuclear dipolar coupling of $\delta_{CH}/(2\pi)$=-45.3 kHz in c) around $\nu_1 = 10$ kHz, $\tau_{90} = 25$ µs, in d) around $\nu_1 = 25$ kHz, $\tau_{90} = 10$ µs.

6b and d) with the line hight plotted as a function of pulse length ($\tau_{90}$ and rf-field amplitude ($\nu_1$). The analytical calculations (top row, of Fig. 6a and b) as well as the numerical simulations of a two-spin system with only a dipolar coupling (second row, Fig. 6c and d) include only effects originating from resonance conditions. The two regions of good decoupling are quite stable against timing errors (typically small fluctuations in the MAS frequency) and also against changes in the rf-field amplitude.

The agreement between the analytical and numerical simulations also show that the analytical calculations characterize the resonance conditions very well. They also show that both areas of interest for good decoupling are fairly insensitive to mis adjustment of the rf-field amplitude but quite sensitive to timing errors. However, timing is very accurate on modern NMR spectrometers and also the MAS frequency is typically stable to a few 10 Hz even at spinning frequencies of 100 kHz.

A more realistic assessment of the decoupling performance in these two regions needs to include also second-order effects

that determine the residual line broadening outside the resonance conditions. To assess this, we have also performed numerical simulations in a $CH_2$ spin system including all dipolar couplings, $J$ couplings, and CSA tensors shown in Fig. 7. While this



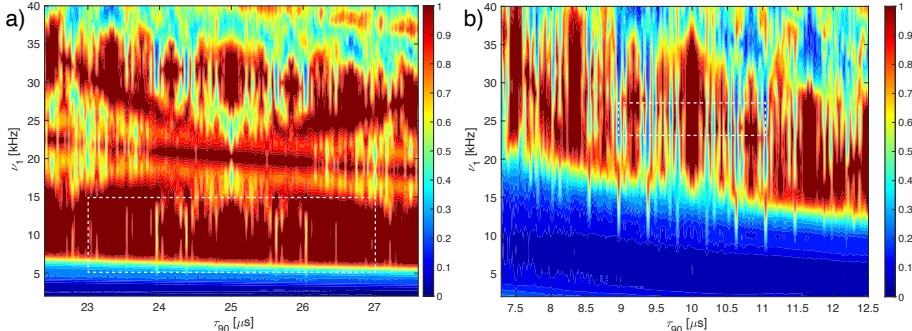

**Figure 7.** Plot of the line intensities in a three-spin $CH_2$ system as a function of the pulse length $\tau_{90}$ and the rf-field amplitude $\nu_1$ for the WALTZ-64 pulse sequence assuming ideal rectangular pulses around the conditions $\omega_r/\omega_1 = 10$ (left side) and $\omega_r/\omega_1 = 4$ (right side). The line height of the numerical simulations with all interactions is shown in a) around $\nu_1 = 10$ kHz, $\tau_{90} = 25$ µs, in b) around $\nu_1 = 25$ kHz, $\tau_{90} = 10$ µs. The parameters used for the simulations can be found in the SI section S1. The white dashed box indicates the area which is covered by the experimental data in Fig. 8

is still a small spin system, all possible interactions are present and all possible second-order terms contribute in such a spin system. The numerical values used in the simulations are given in the SI, section S1. Comparing the three-spin simulations with the two-spin simulations of Fig. 6c and d shows that for small rf-field amplitudes (below about 7 kHz and 15 kHz, respectively)

the peak height becomes much lower, i.e., the lines are much broader. This is due to the inclusion of the heteronuclear $J$ coupling which is only incompletely decoupled by the WALTZ sequence if the flip angles are much smaller than the theoretical values. Otherwise, similar features are visible but they are much more smeared out than in the two-spin calculations of Fig. 6 due to the additional line broadening by second-order terms. It is obvious from these simulations that the timing of the pulses is the most important criterion for setting up low-power WALTZ decoupling to avoid the many resonance conditions especially at

higher $B_1$ fields (shorter pulses) as can be seen in Fig. 7b. Since pulse timing is very accurate even under typical experimental conditions and fluctuations in the spinning frequency are typically below 10 Hz, this is not a critical condition but easy to fulfill. The sensitivity to the exact value of the $B_1$ field is much less pronounced which reflects the good compensation of $B_1$ field deviations from the ideal value of the WALTZ-type sequences (Shaka et al., 1983a, b; Shaka and Keeler, 1987).

Experimental measurements on the pulse-length and field dependence are shown in Fig. 8. The areas covered in the exper-

imental data are much smaller than the ones in the simulations of Fig. 7 to reduce the amount of measurement time required. The experimentally covered area is marked by a dashed white box in the numerical simulations of Fig. 7. There is quite a good agreement between the simulations in the $CH_2$ three-spin system (Fig. 7) and the experimental measurements on the $CH_2$ group of glycine ethyl ester (Fig. 8). The resonance conditions appear at the same pulse length and also the dependence on the rf-field amplitude is very similar. The optimum for the decoupling in the experimental data is shifted towards slightly higher

$B_1$ fields which could be due to inaccuracies in the rf-field calibration or due to $B_1$-field inhomogeneity in the probe.





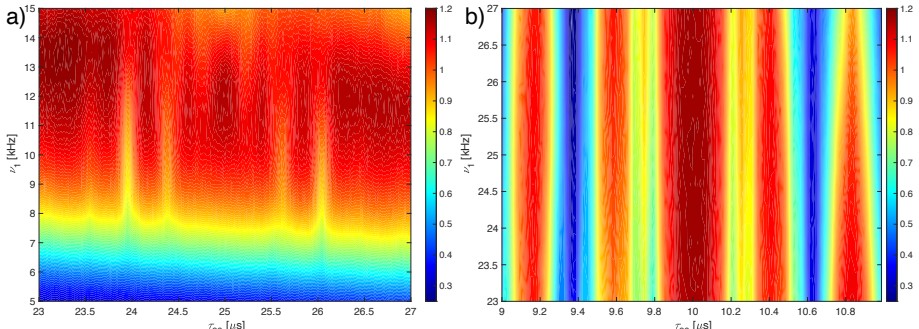

**Figure 8.** Plot of the experimentally measured line intensities as a function of the pulse length $\tau_{90}$ and the rf-field amplitude $\nu_1$ for the WALTZ-64 pulse sequence in glycine ethyl ester around the conditions $\omega_r/\omega_1 = 10$ (left side) and $\omega_r/\omega_1 = 4$ (right side). The line height of the $CH_2$ group in a) around $\nu_1 = 10$ kHz, $\tau_{90} = 25$ µs, in b) around $\nu_1 = 25$ kHz, $\tau_{90} = 10$ µs.

### 3.3 Stability against chemical-shift offset

For high-power decoupling sequences, the effect of chemical shift offsets are often irrelevant because the rf-field amplitudes are more than an order of magnitude larger than typical proton chemical-shift ranges of 10 ppm corresponding to 10 kHz on a 1 GHz spectrometer. In low-power decoupling with rf-field amplitudes typically $\nu_1 < \nu_r/3$, the situation is different

and stability of decoupling with respect to chemical shifts becomes quite important. The WALTZ sequences were designed as broadband rotation pulses which should provide good chemical-shift offset compensation. The experimentally measured dependence on the chemical-shift offset is shown in Fig. 9 for a pulse length of $\tau_{90} = 25$ µs (Fig. 9a) and $\tau_{90} = 10$ µs (Fig. 9b). There are clear modulations of the line intensity over the covered range of chemical shifts but at the optimum $B_1$ field value, the line height never drops below 95% of the high-power reference value for a range of $\pm 10$ kHz for a rf-field amplitude of

10 kHz (Fig. 9a) while for a rf-field amplitude of 25 kHz the dependence is stronger and the line height reaches about 85% of the high-power reference value for an offset of $\pm 5$ kHz (Fig. 9b).

To understand the chemical-shift offset dependence better, numerical simulations in two- and three-spin systems were used. Simulations in two-spin systems with and without $J$ couplings and CSA tensors showed almost perfect decoupling over a large range of chemical shifts and $B_1$ field strengths. This indicates the deterioration of the decoupling performance with increasing

chemical-shift offset as observed experimentally (Fig. 9) is not due to shifts in the resonance conditions due to offset effects. Simulations in a three-spin $CH_2$ system including all interactions (dipolar couplings, $J$ couplings and CSA tensors) are shown in Fig. 10 and show similar characteristics as the experimental data of Fig. 9. This is a clear indication that the deterioration of the decoupling quality with increasing offset is due to cross terms between the homonuclear and heteronuclear dipolar couplings. Indeed, numerical simulations with the homonuclear dipolar coupling set to zero give almost perfect decoupling

over the whole parameter range (see SI, Figs. S12 and S13). The numerical simulations in small spin systems show a stronger offset dependence than observed in experimental data. The source of this discrepancy is not yet fully understood but could be





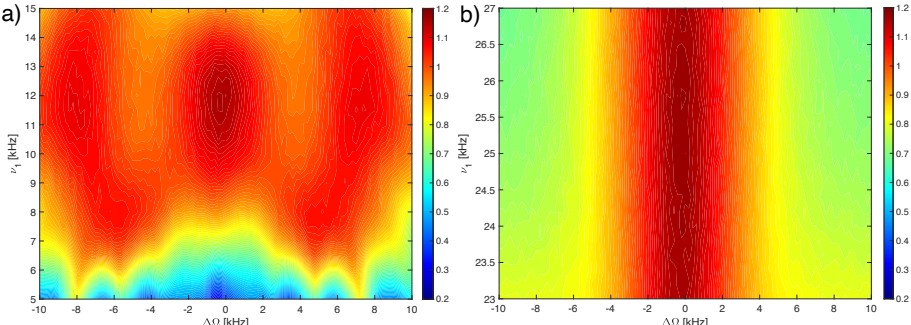

**Figure 9.** Plot of the experimentally measured line intensities as a function of the irradiation offset on the protons ($\Delta\Omega$) and the rf-field amplitude $\nu_1$ for the WALTZ-64 pulse sequence in glycine ethyl ester around the conditions a) $\omega_r/\omega_1 = 10$ corresponding to $\tau_{90} = 25$ µs around $\nu_1 = 10$ kHz and b) $\omega_r/\omega_1 = 4$ corresponding to $\tau_{90} = 10$ µs around $\nu_1 = 25$ kHz.

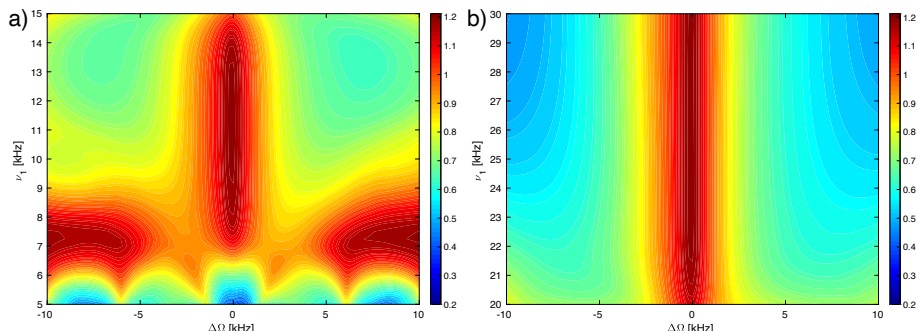

**Figure 10.** Plot of the numerically simulated line intensities as a function of the irradiation offset on the protons ($\Delta\Omega$) and the rf-field amplitude $\nu_1$ for the WALTZ-64 pulse sequence in three-spin $CH_2$ system around the conditions a) $\omega_r/\omega_1 = 10$ corresponding to $\tau_{90} = 25$ µs around $\nu_1 = 10$ kHz and b) $\omega_r/\omega_1 = 4$ corresponding to $\tau_{90} = 10$ µs around $\nu_1 = 25$ kHz.

a consequence of self decoupling of the residual splitting (Sinning et al., 1976; Mehring and Sinning, 1977; Ernst et al., 1998) in large homonuclear-coupled spin systems.

## 4 Conclusions

Low-power decoupling under MAS using WALTZ sequences requires a timing of the sequence that avoids the many potential resonance conditions even though the modulation frequency of the WALTZ sequences in the low-power regime is quite small and very high Fourier coefficients are responsible for the first-order recoupling terms. Two such conditions which were characterized in detail are given at (i) $\nu_1 = \nu_r/10$ leading to a pulse length for the 90° pulse in the WALTZ sequence of $\tau_{90} = \frac{5}{2}\tau_r$ and (ii) $\nu_1 = \nu_r/4$ leading to a pulse length for the 90° pulse in the WALTZ sequence of $\tau_{90} = \tau_r$. There are potentially many more such conditions that could be used for good low-power decoupling using WALTZ sequences. To set up WALTZ decou-





pling, one should first select the pulse length as this is the most important parameter and then do a course optimization of the rf-field amplitude. For samples with low signal-to-noise ratio, the second step can be omitted and the rf-field amplitude estimated from a pulse length determination since the precise setting of the $B_1$ field seems to be not so critical. This makes the WALTZ sequences easy to setup since there is no critical parameter that has to be optimized and the resonance conditions can

be predicted very reliably.

The choice whether WALTZ-16 or WALTZ-64 is used as the decoupling sequence makes little difference as one can see from the simulations and experimental data collected here and shown in the main paper for WALTZ-64 and WALTZ-16 in the Supplementary Information. At first glance, it might look like the WALTZ-16 sequence has narrower areas where the Fourier coefficients are zero but the fewer numbers are compensated by the factor four in spacing of the resonance conditions. The

very similar behavior of WALTZ-16 and WALTZ-64 with respect to resonance conditions can be seen from Fig. 11 where we plot the simulated peak heights in a $CH_2$ group including all interactions for WLATZ-64 and WALTZ-16 but with the same time resolution of the $\tau_{90}$ pulse length. This constitutes an oversampling of the WALTZ-16 sequence by a factor of four compared to the sampling at only the resonance condition as shown in Fig. S3. The WALTZ-64 peak heights are marked by a blue circle, the WALTZ-16 peak heights by a red triangle and the points belonging to resonance conditions of WALTZ-16

by red hexagons. One can clearly see that the width of the resonance conditions is very similar and that there is no clear advantage of WALTZ-64 over WALTZ-16 in the width of the areas where we have no residual line broadening from resonance conditions. The similarities between Waltz-16 and WALTZ-64 decoupling can also be seen from Fig. S16 in the SI where the experimental peak height for both decoupling sequences are shown in a single plot as a function of the pulse length. There are small differences visible between the two curves but there are no significant differences in the areas where we expect good

decoupling.

To set up low-power WALTZ decoupling, it is recommended to use the $\nu_1 = \nu_r/10$ for 100 kHz MAS frequencies and beyond (0.7 mm rotors and smaller). For 50-60 kHz MAS (1.3 mm rotors), the $\nu_1 = \nu_r/4$ condition is most likely better due to the higher rf-field amplitude. Where the transition from one condition to the other exactly happens, will also depend on sample composition especially the proton density of the sample. To answer this question more experimental data on different samples

and spinning frequencies is needed.

How the WALTZ sequences compare to other commonly-used low-power decoupling sequences like AM-XiX (Agarwal et al., 2013), SWf-TPPM (Thakur et al., 2006), SPINAL-64 (Garg et al., 2024), or ROSPAC (Simion et al., 2022) under different experimental conditions, i.e., spinning frequency range and proton density, is an important question that is beyond the scope of this paper. This topic is currently under investigation in our laboratory.

*Code and data availability.* The experimental data, the code to generate the analytical calculations, the numerical simulations, and the processing scripts to generate the figures will be made available through the ETH Research Collection after acceptance of the paper.





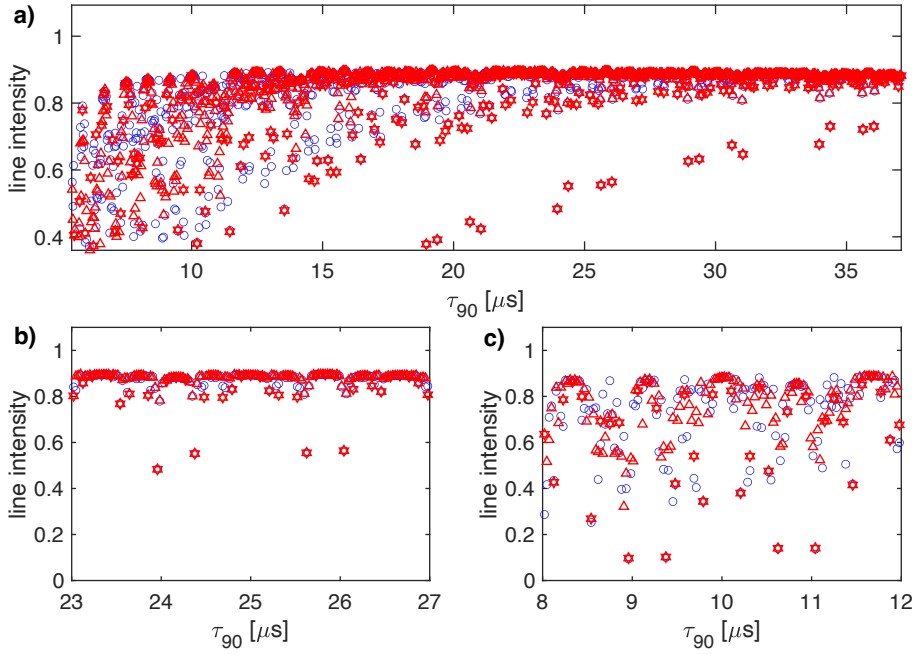

**Figure 11.** Plot of the simulated line intensity in a $CH_2$ three-spin system with dipolar couplings and CSA tensors as a function of the pulse length $\tau_{90}$ for the WALTZ-64 and the four times oversampled WALTZ-16 pulse sequence assuming ideal rectangular pulses. The WALTZ-64 peak heights are marked by blue circles, the WALTZ-16 peak heights by red triangles and the points belonging to resonance conditions of WALTZ-16 by red hexagons. The corresponding rf-field amplitude for each value of $\tau_{90}$ can be calculated as $\nu_1 = 1/(4\tau_{90})$. The pulse length correspond to values of $\tau_{90} = k_0 \tau_r/(4*z_0)$ a) Complete range from $\tau_{90} = 2.5$ to 50 µs corresponding to $k_0 = 96$ to 1920. b) Enlarged range around $\tau_{90} = 25$ µs corresponding to an ideal $B_1$ field of 10 kHz. c) Enlarged range around $\tau_{90} = 10$ µs corresponding to an ideal $B_1$ field of 25 kHz. As described in the materials and methods section, all simulations are processed with an exponential apodization of 30 Hz with an intensity of one corresponding to a non decaying line.

*Author contributions.* ME designed the research, LT and ME implemented and analyzed the simulations and the measurements. ME wrote the first paper draft and both authors edited and contributed to the writing process to generate the final manuscript.

*Competing interests.* At least one of the (co-)authors is a member of the editorial board of Magnetic Resonance. The authors have no other competing interests to declare.

*Acknowledgements.* We would like to thank Paul Schanda for insightful discussions about low-power decoupling and Kathrin Aebischer for the simulation parameters. Rainer Kümmerle (Bruker AG) is acknowledged for providing access to a beta release of Topspin 4.4 which al-




lowed higher time resolution for the decoupling pulse length. ME acknowledges support by the Schweizerischer Nationalfonds zur Förderung der Wissenschaftlichen Forschung (grant no. 200020_219375).



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
