# Peer review of "Low-Power WALTZ Decoupling under Magic-Angle Spinning"

_Magnetic Resonance, 2024_

## Author Response (AR2)

**There are no changes to the previous version since that contained already all the changes requested by the reviewers.**

**Reviewer 1:**

The manuscript from Thomas and Ernst explores the theoretical basis of heteronuclear solution-state decoupling sequences such as WALTZ-16 and WALTZ-64 under fast MAS and low-rf conditions. Floquet theory and numerical simulations have been employed to understand the contribution of the recoupling terms at different rotary-resonance conditions and identify best decoupling conditions. The manuscript is well formulated, clearly identifies and elaborates the best decoupling conditions at rf strengths of 0.1 and 0.25 times the spinning frequencies. In principle the manuscript should be accepted. I only have some minor comments.

1. In figure 1c, for k0 ~1950 why do the coefficients become negative. Can az(k0) be really negative?

2. A critical reference and comparisons concerning WALTZ decoupling at fast MAS by Ishii and co-workers is not cited (solid-state NMR vol 72, page 9-16, 2015). The authors of this paper argue that the pulse length is very critical and not so much the rf field, mainly due the resonance conditions. In contrast, Ishii and co-worker argue that at constant rf field the flip angle (Figure 8) is not so relevant or in other words the pulse length is not a relevant parameter for flip angles between 55-360 degrees. How does one explain these different conclusions.

3. Ishii and co-workers also show that WALTZ-16 works rather well even at 1H rf amplitude of 5 kHz. This is in contrast to the simulations shown in Figure 7 and the SI figure of this manuscript. How does one explain this? Could the difference originate from employing different chemical moieties in the analysis: This manuscript uses -CH2 groups while Ishii's publication mainly uses CH3/CH.

4. There are many simulations in the manuscript which is essential but can become very confusing to the reader. I feel explicitly stating the interactions considered in the figure can help reduce the confusion and constant browsing through the text looking for what interaction is considered for the particular figure.

5. Figure 2: two spin-interaction, with only heteronuclear dipolar coupling included. Why is the J coupling left out. Refocusing of J coupling will happen at only selective flip angles, wouldn't lack of J coupling overstate the performance of WALTZ at conditions where J coupling is not refocused.

6. Figure 9a and 10a: at off-resonance conditions why do numerical simulations and experiments differ? For experiments showing higher intensities at off-resonance conditions one could argue that normalization factor for experiments is lower due to

various reasons. However, around ~7 kHz rf the numerical simulations show enhanced intensities compared to rf field of 11-14 kHz. In contrast the experiments show an opposite trend. Is there a reasonable explanation.?

7. Composite-pulse sequences do not work in the slow MAS and high rf field regime. In the conclusions the authors could possibly comment why composite pulse decoupling does not work in the slow MAS and high rf field regime at typical ratio of nu_1=10*nu_r. This is however an optional exercise.

**Response:**

Thank you for the positive assessment and especially for pointing out the work by Ishii which we were not aware of and which did not show up in a literature search. Here is a point-by-point answer to the questions and suggestions:

1. The negative points in Fig. 1c are a mistake generated when assembling the varies sub figures from the Matlab plots. The Figure has been corrected and the reviewer is correct that the a_z^(k_0) all have to be larger/equal zero. You can also see this from the shifted axis on the right hand side. Thanks for pointing this out, we should have noticed this ourselves.

2. Thanks for pointing out this experimental study. We have added the reference to Ishii's paper in the introduction: "There is only a single experimental study \citep{Wickramasinghe:2015fl} where the properties of WALTZ-16 decoupling have been investigated and compared to other low-power decoupling sequences. However, it is not clear what the best parameters for using low-power WALTZ sequences at a given spinning frequency are."
The results by Ishii are consistent with ours. He states that he did a local optimization of the pulse length for each of the rf-field amplitudes which in essence amounts to avoiding the resonance conditions that we discuss extensively. I think it is not surprising that the WALTZ sequences are fairly forgiving towards the flip angle of the pulse as long as the timing is not changed. This is in agreement with our results (Figs. 7 and 8) that show a broad range of good decoupling in B1. It is not clear from Ishii's paper whether they changed the pulse length or the B1 field. Even if they changed the pulse length, at a ratio of 1:20, the resonance conditions become quite weak.

3. We never looked in detail at a rf-amplitude as low as 5kHz with the corresponding pulse length of 50us. Figures 7 and 8 show decoupling around a pulse length of 25us with an rf-field amplitude as low as 5 kHz which is of course not optimal. Fig. 4 shows that decoupling with 5 khz/50us experimentally works but the lower the $B_1$ field, the lower the line intensity gets and we did not investigate this area in more detail. So I think there is no contradiction to the results of Ishii.

4. We have made a little icon that shows which interactions are present in the various numerical simulations. We hope that adding these icons to the plots of numerical simulations will help in a faster understanding of the plots.

5. The restriction in Fig. 2 to only dipolar couplings (omitting J couplings) was intentional to allow a better comparison to Fig. 1. However, since Fig.2 was always run at the perfect flip angle, the J couplings would be refocused in this plot. The influence of J couplings can best be seen when comparing Figs. 6 and 7. The bad decoupling in Fig. 8 at too small flip angles (B1 lower than it should theoretically be) which is not present in Fig. 7 where the J coupling is omitted.

6. We are not fully sure why the simulations and the experimental results of the off-resonance decoupling show significant differences. This is already mentioned in the text (line 255): "The numerical simulations in small spin systems show a stronger offset dependence than observed in experimental data. The source of this discrepancy is not yet fully understood but could be a consequence of self decoupling of the residual splitting \citep{Sinning:1976wc, Mehring:1977tl, Ernst:1998vl} in large homonuclear-coupled spin systems." But so far this is speculation.

7. This is an interesting question. Experimentally, we do not have data since the maximum B1 field that we measure was 100 kHz as can be seen from Fig. 4. From the resonance conditions, there are areas with $\omega_1/\omega_r > 1$ that in theory allow high-power decoupling, for example $\omega_1/\omega_r$ = 24, 12, 6 or 3. We have not done any simulations in this regime so we do not know how well this would work. My feeling is that the adjacent resonance conditions become very strong and will make decoupling in this regime not very efficient. We will try to look into this and see whether we can say something about the possibility to use WALTZ sequences under high-power decoupling conditions.

**Reviewer 2:**

This is a timely manuscript, as the WALTZ sequence is increasingly becoming a popular the sequence for 1H decoupling under fast MAS frequencies. The manuscript is clearly written, and convincingly demonstrates that robust decoupling is to be expected with this sequence on theoretical grounds at an rf of 10 kHz (at ~100 kHz MAS). Experimental data justify these claims. This article also gives data to estimate the robustness of the sequence over varying parameters (offset, flip angle, rf). This is especially important as the sequence is expected to work a bit differently than it does in solution NMR due to various factors. I recommend that this manuscript be published as is. I have a couple of minor points that the authors should may address in a final version:

1. The article uses a standard non-deuterated system for experimental measurements, and the theory is also given for spin-system that is appropriate for this. However, although it is true that quite a few articles have demonstrated the utility of WALTZ decoupling at fast MAS in non-deuterated systems, it is more often the sequence of choice in deuterated (and back-exchanged) systems, and which may have a significantly different behaviour. I agree that the present article is self-consistent, but it would perhaps be useful to mention this point somewhere in the introduction.

2. The condition chosen here is nu_m = 416.66Hz and nu_r = 100 kHz, which is technically a resonance condition (admittedly for very large k values, as has been described in several figures). Do the authors expect a slightly better performance if the nu_m is shifted slightly away from this condition, which can push the resonance conditions to even higher value of k? (for example 10.5 or 11 kHz at 100 kHz MAS?). According to Fig 4 (b), this does not seem to be so as the intensity at ~23.5 us does seem to be very very slightly higher than 25us. Of course, this could also be the effect of a slightly higher rf, but would it be prudent to use a slightly different condition that is not exactly 10 kHz?

**Response:**

We would like to thank the reviewer for the positive words about the manuscript.

1. This is an important point and we have added a paragraph to the introduction discussing the two different situations: "There are mainly two regimes where low-power decoupling is currently used: (i) in fully protonated systems spinning at around 100 kHz and higher (using 0.7 mm outer-diameter rotors or smaller); (ii) in deuterated and back-exchanged systems spinning around 60 kHz (using 1.3 mm outer-diameter rotors). We focus on the properties of WALTZ sequences under the first condition since decoupling in fully-protonated systems is more demanding due to the strong homonuclear proton-proton couplings that are absent in deuterated and back-exchanged systems." We also added the two different cases explicitly to the discussion in the Conclusions. As stated in the conclusions a detailed comparison of the performance of different sequences under these conditions is beyond the scope of the paper.

2. This is a difficult point to answer since the spacing of the resonance conditions is quite dense. At 100 kHz MAS two resonance conditions are only separated by 0.02604 us for WALTZ-64 and by 0.1042 us for WALTZ-16. while the spacing is linear in the pulse length it is non linear in the rf-field amplitude but the difference in rf-field amplitude between two resonance conditions around 10 kHz is around 10 Hz. The width of the resonance conditions is most likely larger than the spacing. Simulations of the WALTZ-16 sequence with oversampling in between the resonance conditions show that these points are not zero but show residual broadening from nearby resonance conditions. Therefore, it is unlikely that making the sequence actively asynchronous will have a visible effect on the line width. In addition, as the reviewer points out, such effects were also not observed in the experimental data where the timing increment was 12.5 ns. But we should keep in mind that the experimental data always show a convolution with the rf-field inhomogeneity which might also add some additional line broadening.

Since we have already indicated the changes made to the manuscript in the two online responses, we do not feel that additional comments are required. We hope that these changes make the manuscript suitable for publication in MR.